# Influence of Ventilation Openings on the Energy Efficiency of Metal Frame Modular Constructions in Brazil Using BIM

**Mohammad K. Najjar** [1,2,*], **Luis Otávio Cocito De Araujo** [1], **Olubimbola Oladimeji** [3], **Mohammad Khalas** [4], **Karoline V. Figueiredo** [2], **Dieter Boer** [5], **Carlos A. P. Soares** [6] **and Assed Haddad** [1,2]

1   Departamento de Construção Civil, Universidade Federal do Rio de Janeiro, Rio de Janeiro 21941-853, Brazil
2   Programa de Engenharia Ambiental, Universidade Federal do Rio de Janeiro, Rio de Janeiro 21941-853, Brazil; karolinefigueiredo@poli.ufrj.br
3   Department of Building, Osun State University, Osogbo 210001, Nigeria; oladimeji70@yahoo.co.uk
4   Department of Design, Faculty of Architecture, Tishreen University, Lattakia P.O. Box 2230, Syria; dr.arch.mk@gmail.com
5   Departament d'Enginyeria Mecanica, URV, Universitat Rovira i Virgili, 43003 Tarragona, Spain; dieter.boer@urv.cat
6   Pós-Graduação em Engenharia Civil, Universidade Federal Fluminense, Niterói 24020-141, Brazil; capsoares@id.uff.br
*   Correspondence: mnajjar@poli.ufrj.br

**Abstract:** Construction projects demand a higher amount of energy predominantly for heating, ventilation, and illumination purposes. Modular construction has come into the limelight in recent years as a construction method that uses sustainable building materials and optimizes energy efficiency. Ventilation openings in buildings are designed to facilitate air circulation by naturally driven ventilation and could aid in reducing energy consumption in construction projects. However, a knowledge gap makes it difficult to propose the best dimensions of ventilation openings in buildings. Hence, the aim of this work is to empower the decision-making process in terms of proposing the best ventilation opening dimensions toward sustainable energy use and management in buildings. A novel framework is presented herein to evaluate the impact and propose the best dimensions of ventilation openings for metal frame modular construction in Brazil, using building information modeling. The ventilation openings were constructed and their dimensions evaluated in eight Brazilian cities, based on the bioclimatic zone (BioZ) classification indicated in ABNT NBR 15220: Curitiba (1st BioZ); Rio Negro (2nd BioZ); São Paulo (3rd BioZ); Brasília (4th BioZ); Campos (5th BioZ); Paranaíbe (6th BioZ); Goiás (7th BioZ); and Rio de Janeiro (8th BioZ). The study results show that the energy consumption of the same building model would vary based on the dimensions of ventilation openings for each BioZ in Brazil. For instance, modeling the same modular construction unit in the city of Rio Negro could consume around 50% of the energy compared to the same unit constructed in the city of Rio de Janeiro, using the small opening sizes based on the smallest dimensions of the ventilation openings. Similarly, modeling the construction unit in Curitiba, São Paulo, Brasília, Campos, Paranaíba, and Goiás could reduce energy consumption by around 40%, 34%, 36%, 18%, 20%, and 16%, respectively, compared to constructing the same building in the city of Rio de Janeiro, using the small opening sizes based on the smallest dimensions of the ventilation openings. This work could help practitioners and professionals in modular construction projects to design the best dimensions of the ventilation openings based on each BioZ towards increasing energy efficiency and sustainability.

**Keywords:** modular construction; building information modeling; ventilation openings; energy efficiency



## 1. Introduction

The construction sector requires the use of various equipment and appliances over its lifespan to extract raw materials, and for fabrication, transportation, construction, operations, and demolition [1]. Hence, this sector has high energy demands and is known as

"the industry of the 40%" [2], with an expectation that this figure will double by 2050 [3]. It is responsible for almost 40% of the global $CO_2$ emissions and natural resource consumption [4,5]. Energy consumption in the construction industry can be assessed based on the relevant contained energy [6]; embodied energy, which refers to evaluating the amount of energy consumed for the extraction of raw materials and fabrication of building materials; grey energy and induced energy, which refer to the energy consumed during the transportation process of materials to the construction site and the energy consumed during the construction period, respectively; operating energy, which is the energy consumed over the occupation period of buildings; and disposal energy, which is the energy used for the demolition purposes. This work focuses on the operating energy in buildings, where the use of appliances for heating, ventilation, and illumination purposes could influence this type of energy [7]. Several factors must be considered when analyzing this fact, such as the construction materials, ventilation opening dimensions, shadings, and building orientation [8]. At this level of ventilation openings, operating energy considers the major role of windows as the link between the external and internal environments, with great importance for energy performance and thermal comfort [9]. Ventilation openings provide natural light for the interior environment and are responsible for natural ventilation and passive cooling [10]. Effective assessment and design of the ventilation openings in construction projects could lessen the necessity of using the heating, ventilation, and air conditioning (HVAC) systems and, consequently, reduce the energy consumption in buildings [11].

In the literature, several methods and tools have been applied to tackle the energy efficiency issue in construction projects, such as nearly zero-energy buildings (NZEBs) [12] and building information modeling (BIM) [13]. There are several modeling and optimization tools to evaluate the energy management and efficiency in buildings that use BIM combined with other tools such as sustainability [13], life cycle assessment (LCA) [14], life cycle cost (LCC) [15], building energy modeling (BEM) methodology [16], and multi-criteria decision-making (MCDM) [17]. BIM tools can be applied at different stages of the building, whether in the construction project [18] or during its useful life [19]. Moreover, novel construction processes have been developed to reduce the energy consumption in buildings, such as modular construction, which appears as an important step toward producing more sustainable and energy-efficient buildings [20], and reducing the greenhouse gas emissions and carbon embodied in buildings [21]. Modular construction applies the principle of constructive design using modules prefabricated off-site and assembled in situ, reducing construction time, waste, and cost; it also lessens vulnerability to stoppages in construction due to weather conditions, and makes easier the choice of materials as well as deconstruction at the end of its useful life, reducing the need for manual steps and labor [22]. Several construction materials can be applied in the modular construction process, such as wood, metal, and concrete [23]. For example, a mixture of wood and light-gauge metal is being used for low-rise buildings, while concrete and hot-rolled steel are being used for skyscrapers [24]. The application of metal structures in modular construction results in shorter installations, resistance to weather conditions, less load on the foundation, easy adaptation to other uses, and high resistance to external factors when applying galvanized steel [25].

In the literature, several studies have examined the important role of ventilation openings in increasing the energy efficiency of buildings [26–30]; however, a knowledge gap makes it challenging to propose the best dimensions of the ventilation openings of construction projects for more sustainable and energy-efficient buildings. Thus, this work aims to empower the decision-making process in terms of proposing the best ventilation opening dimensions for sustainable energy use and management in buildings. A novel framework is presented herein to evaluate the impact and propose the best dimensions of ventilation openings for metal frame modular constructions in several Brazilian bioclimatic zones (BioZ), using BIM. This work aims to help practitioners and professionals working on modular construction projects in Brazil to design the best dimensions of ventilation openings based on each BioZ. The evaluation of energy performance in modular construction, considering the annual energy consumption and the yearly energy end use for HVAC and lighting purposes in buildings,

is conducted to design and simulate the dimensions of ventilation openings in buildings. BIM tools are applied to assess the energy performance of the buildings based on the various opening dimensions. Hence, Autodesk Revit is used to simulate a case study of the modular construction unit, and Autodesk Green Building Studio is used to simulate the designed sequences of the proposed ventilation opening values based on the Brazilian standard (ABNT NBR 15220) of several Brazilian bioclimatic zones.

This paper is organized as follows: in Section 2, a comparative background on modular construction and bioclimatic zones of Brazil, as well as the impact of ventilation openings on energy efficiency in buildings, is presented, while in Section 3, the materials and methods applied in this work are described. Next, in Section 4, a case study of a modular construction unit is used to validate the components of the proposed framework. Then, the results of this work are presented and discussed in Section 5.

## 2. Literature Review

This part of the study has been divided into three parts. The first highlights the modular construction approach. This type of construction is the focus of the study as it has been under development in recent years in Brazil. The second part discusses the impact of ventilation openings on energy efficiency in buildings, while the last part presents the bioclimatic zones of Brazil.

### 2.1. Modular Construction Design

Modular construction is a developed process that could improve indoor air quality in construction projects for green buildings [31]. Digital technologies are important in enhancing the fabrication process [32] and the prefabrication procedures of such approaches [12]. For instance, BIM tools have been used to automate the manufacturing process of modular construction buildings in China [33] and South Korea [34]. Modular construction necessitates fewer workers on the construction site, reduces the construction time, and, consequently, decreases the construction costs compared to conventional construction [31]. The modular construction approach has been used to reduce the embodied energy in construction, mainly by reducing waste [35]. Generally, a range of unit dimensions is provided in the modular construction market that varies between 2.5 m × 3.6 m, 3 m × 3 m, 3 m × 6 m, 3 m × 9 m, and 3 m × 12 m, to fit any type of construction project [36]. Anderson et al. applied modularity directly in embodied energy techniques, showing significant benefits compared to the traditional method [37]. Hammad et al. carried out the modeling of two modular projects and two projects applying traditional techniques, with more advantageous results observed for the modular construction concerning the energy used during the construction process and embodied energy, both by the materials and by the constructive elements [38]. Tavares and Freire combined BIM and life cycle assessment for prefabricated steel houses in seven different locations, with three levels of insulation and two heat pumps, showing less built-in impact but operational impacts similar to conventional constructions [39].

The selection process of building materials is a crucial point in the analysis of modular construction, where each component presents specifications according to the end use. Metal frame modular construction is widely used for residential, commercial, and industrial purposes. The system has the benefits of durability, fire resistance, water tightness, acoustic insulation, flexibility, and lightness, reducing the weight restrictions and overlaps of traditional modular constructions [40]. Wood frame modular construction has high strength and durability, as well as low density, allowing its use in multi-story buildings with maximum height and slenderness as limiting factors [41], service life [42], and outdoor modular structures [43]. Concrete is widely used in modular high-rise buildings due to its greater load-bearing capacity [44]. Moreover, recent developments in the modular construction market have highlighted some increasingly advanced materials, such as geopolymers (i.e., steel–geopolymer concrete composite [45] and geopolymer concrete combined with fly ash and blast furnace slag [46]).

## 2.2. Impact of Ventilation Openings on Energy Efficiency in Buildings

Ventilation openings have a major role in determining the energy efficiency and thermal comfort of buildings [47,48]. Salcido et al. assessed the mixed ventilation systems for the integration of natural ventilation in office buildings, where, concerning windows, the wind behavior is fundamental for their design and controlling their opening period [49]. Lotfabadi and Hançer [50] developed a novel multi-objective optimization model of building openings to evaluate the visual comfort and energy efficiency of an office building in Cyprus. The authors highlighted that the window-to-wall ratio and glazing type could play a major role in energy efficiency optimization in buildings. The output results illustrated that a window-to-wall ratio between 20% and 50% could increase the energy efficiency of the case study building in Cyprus. Tien et al. [28] presented a vision-based deep learning framework for the detection and recognition of window operation toward effective management of ventilation heat losses in buildings. The authors confirmed that the proposed framework could increase energy efficiency in buildings through the detection of the dimension and types of windows. Matour et al. [51] evaluated the impact of opening configurations on the ventilation performance of an integrated tall building with a double skin façade, taking into consideration the major four wind orientations of $0°$, $30°$, $60°$, and $90°$. The evaluation process was conducted based on three ventilation performance indicators, namely, induced air flow, wind speed ratio, and airflow distribution across the cavity length. Bauer et al. [52] analyzed the household practices and the construction components of the buildings and observed that the positioning of windows could play a major role in assessing the energy efficiency of construction projects, through the direct and indirect loss of heating energy in buildings. Dussault et al. analyzed different levels of window opacity and economy in the HVAC system, beneficial in temperate climates, taking into account the window direction [9]. Fabi et al. identified and analyzed the driving forces for the behavior of opening and closing windows, including physiological, psychological, social, physical, and contextual environments, and how these factors could influence ventilation and thermal control in residential buildings and offices [53].

Furthermore, BIM tools have been applied in the literature to assess the role of ventilation openings in the energy efficiency in buildings. For instance, Kim et al. [54] examined 65 different design scenarios based on the size, position, and orientation of windows in buildings, using Autodesk Green Building Studio as a BIM tool. The authors found that energy consumption in buildings would increase when using high values of window-to-wall ratio, and vice-versa. Gan et al. [55] proposed a BIM-based framework to evaluate the natural ventilation impact on energy efficiency and thermal comfort in buildings. The authors highlighted that BIM tools could facilitate the evaluation process of natural ventilation to maintain indoor thermal comfort and reduce energy consumption in buildings. Zhang et al. [56] prepared a BIM-based architectural analysis to optimize building design and concept. The authors pointed to the importance of BIM in evaluating the effect of construction components in energy-efficient buildings—basically, the external walls and ventilation openings. Similarly, Maglad et al. [57] elaborated a case study building to assess the impact of BIM tools on energy consumption evaluation in buildings, using Autodesk Insight 360 and Green Building Studio. The authors confirmed that the BIM methodology could promote energy efficiency in construction projects.

## 2.3. The Bioclimatic Zones of Brazil

Brazil has two standards regarding thermal comfort in buildings, ABNT NBR 15220 for construction projects [58] and ABNT NBR 15575 for residential building performance [59]. Standard ABNT NBR 15220 is divided into three parts: the first with definitions, symbols, and units, the second referring to calculations, and the third referring to the bioclimatic zones of the country, the latter being used in this study. The third part of this standard establishes Brazilian climate zoning: due to the country's enormous size, it is divided into eight zones, classified based on information on temperature, humidity, and climatological data, and the adaptation of the bioclimatic chart [46]. The size of the ventilation openings,

the protection of the openings, external sealing, and passive thermal conditioning strategies are presented, as well as a list of 330 cities with their climates categorized [58]. The use of the bioclimatic zone (BioZ) classification in the construction sector in Brazil has been examined in several studies. For example, Rabbi and Nico applied simulations to evaluate thermal comfort, energy efficiency, and economic viability in social housing for the eighth BioZ [60]. Kruger and Mori compared the original design of a bank branch and an optimized one in its envelope and observed that changes in the facade led to a reduction in electricity consumption [61].

## 3. Materials and Methods

The aim of this work was to empower the decision-making process in terms of designing the ventilation opening dimensions for sustainable energy use and management in buildings. This work proposes a framework that facilitates evaluating the impact and proposes the best dimensions of ventilation openings for metal frame modular construction in several Brazilian bioclimatic zones (BioZ) using BIM, as presented in Figure 1. Such a framework could help practitioners and professionals in modular construction projects to design the best dimensions of the ventilation opening based on each BioZ, towards increasing energy efficiency and sustainability.

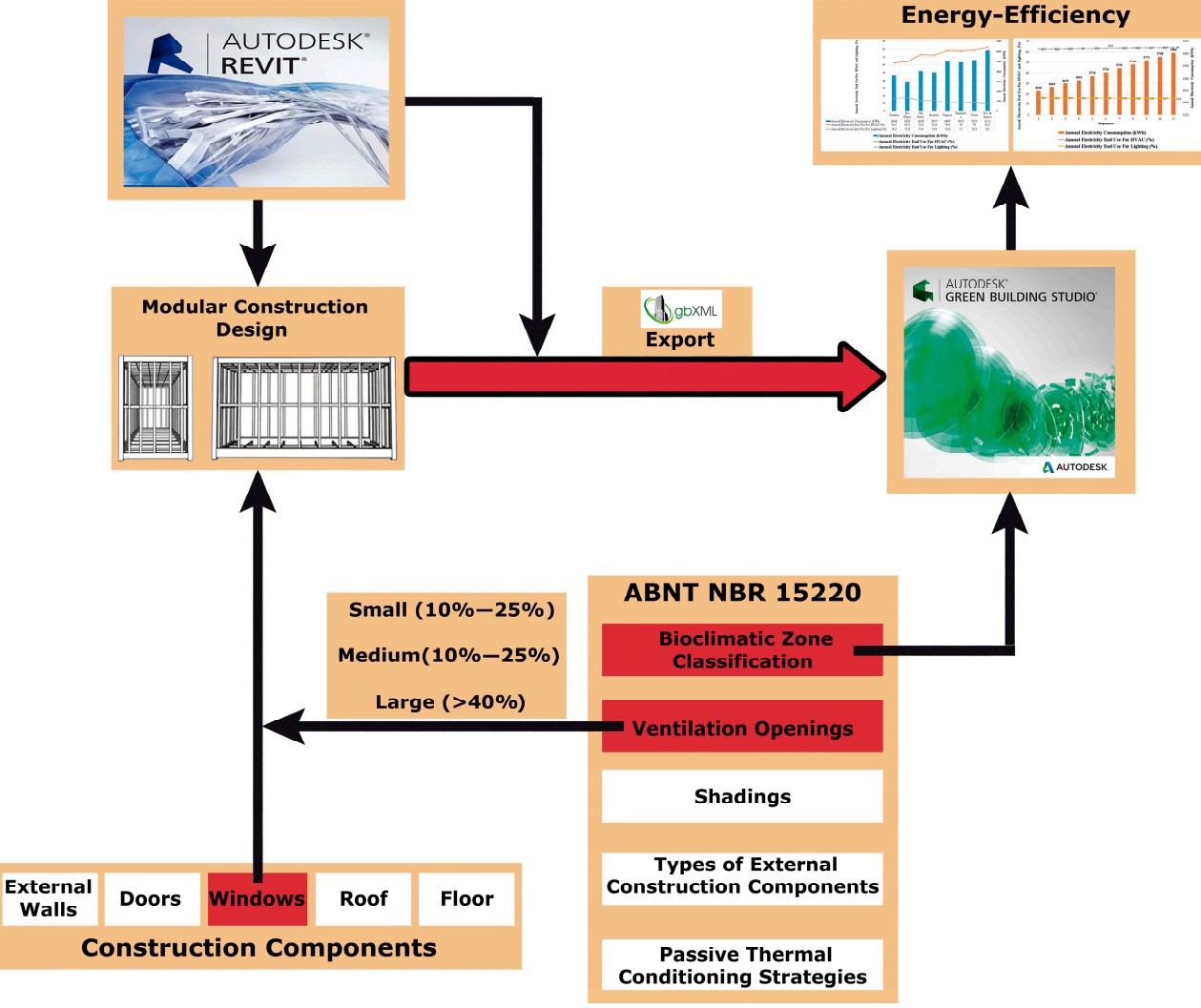

**Figure 1.** The proposed framework.

*3.1. Metal Frame Modular Construction Design*

The preliminary step of the proposed framework is to use Autodesk Revit as a BIM software to conduct the 2D and 3D simulations of the metal frame modular construction building, using the families of construction components of such types of buildings. The simulations take into account the physical and thermal properties of the components of the building envelope (i.e., external walls, doors, windows, roof, and floor), which could enable intelligent building modeling [2].

The next step is to modify the dimensions of the openings based on the recommended ventilation openings presented in ABNT NBR 15220 [58]. This step requires considering the BioZ classification of the Brazilian cities as presented in the third part of ABNT NBR 15220. This standard divides the Brazilian territory into eight different bioclimatic zones, indicating three ranges of ventilation opening dimensions, as follows:

(a)    Medium ventilation openings: a range of dimensions between 15% and 25% of the total floor area of the built environment. These dimensions are recommended in the cities located in the first, second, third, fourth, fifth, and sixth bioclimatic zones.

(b)    Small ventilation openings: a range of dimensions between 10% and 15% of the total floor area of the built environment. These dimensions are recommended in the cities located in the seventh bioclimatic zone.

(c)    Large ventilation openings: a recommended dimension equal to or above 40% of the total floor area of the built environment. These dimensions are recommended in the cities located in the eighth bioclimatic zone.

The focal point of this work is examining the impact of ventilation openings in metal frame modular constructions in light of the recommended dimensions for each BioZ in Brazil. First, the method requires modeling the proposed sequences of the study by defining the location where the building is to be constructed to define the BioZ of the selected city. Then, it is possible to perform the selected range of ventilation opening dimensions individually in Autodesk Revit.

*3.2. Data Inventory*

This step necessitates compiling asset and liability data for the organization of the study. This in turn necessitates exporting the sequences performed in Autodesk Revit into gbXML format to be importable and usable as a new project in Autodesk Green Building Studio, which is applied as a BIM tool herein to simulate and examine the energy efficiency of construction projects [62], as presented in Figure 1. At this level of analysis, there is a need to insert some basic information such as defining the project name, building type (i.e., residential, commercial, educational, health, retail, etc.), occupancy rate (i.e., how many hours per day and days per week the building is to be occupied by users; 24/7 facility, 12/6 facility, etc.), and defining the address of the building, which enables defining the weather station and time zone of the building.

*3.3. Evaluation Process*

After performing all the designed sequences, an evaluation process of the collected results is mandatory based on the annual electricity consumption and annual electricity end use for the lighting and HVAC systems, to achieve the objectives of the study by defining the impact of ventilation openings on the energy efficiency of metal frame modular construction projects. In this work, the evaluated parameters are based on the recommended ventilation opening dimensions for each BioZ in Brazil based on ABNT NBR 15220. It is important to declare that one can implement any other ventilation opening dimensions within several zones. However, the case study in this work is used to validate the accuracy of the proposed framework. The evaluation process can be performed using MS Excel to facilitate the comparison step using graphics.

## 4. Case Study and Simulation Analysis

The case study was modeled based on a modular construction unit simulated in eight cities that are classified within diverse bioclimatic zones [58]: Curitiba in the state of Parana

(1st BioZ); Rio Negro in the state of Parana (2nd BioZ); São Paulo in the state of São Paulo (3rd BioZ); Brasilia in the state of Brasilia (4th BioZ); Campos in the state of Rio de Janeiro (5th BioZ); Paranaíba in the state of Mato Grosso do Sul (6th BioZ); Goiás in the state of Goiás (7th BioZ); and Rio de Janeiro in the state of Rio de Janeiro (8th BioZ) [58]. The important fact is that the selected cities are in different geographical locations in Brazil. The selection of the cities is based on the aim of evaluating the energy efficiency of such types of buildings in all the bioclimatic zones of Brazil.

Part 3 of ABNT NBR 15220 classified and detailed the required bioclimatic strategies (i.e., ventilation openings, shadings, types of external construction components, and passive thermal conditioning strategies) for construction projects in each BioZ of Brazil. This work will examine the requirements for ventilation openings in the selected cities. According to ABNT NBR 15220, three sizes of ventilation openings are indicated, as follows [58]: small (between 10% and 15%) in zone seven; medium (between 15% and 25%) in zones one, two, three, four, five, and six; and large (>40%) in zone eight. This work will consider the values of the ventilation openings in the selected cities based on the relevant BioZ within an increase of 1%, as presented in Table 1. In these terms, in the cities of Curitiba, Rio Negro, São Paulo, Brasília, Campos, and Paranaíba, eleven values of ventilation openings are examined (15%, 16%, 17%, 18%, 19%, 20%, 21%, 22%, 23%, 24%, and 25%). In the city of Goiás, six values of ventilation openings are examined (10%, 11%, 12%, 13%, 14%, and 15%). In Rio de Janeiro, the required value of ventilation openings is more than 40%; hence, this work will examine six values of ventilation openings (40%, 41%, 42%, 43%, 44%, and 45%).

**Table 1.** Bioclimatic zones and the tested ventilation openings for the selected cities.

| BioZ | City | Ventilation Openings |
|:---:|:---:|:---:|
| 1 | Curitiba | Medium (15%, 16%, 17%, 18%, 19%, 20%, 21%, 22%, 23%, 24%, and 25%) |
| 2 | Rio Negro | Medium (15%, 16%, 17%, 18%, 19%, 20%, 21%, 22%, 23%, 24%, and 25%) |
| 3 | São Paulo | Medium (15%, 16%, 17%, 18%, 19%, 20%, 21%, 22%, 23%, 24%, and 25%) |
| 4 | Brasília | Medium (15%, 16%, 17%, 18%, 19%, 20%, 21%, 22%, 23%, 24%, and 25%) |
| 5 | Campos | Medium (15%, 16%, 17%, 18%, 19%, 20%, 21%, 22%, 23%, 24%, and 25%) |
| 6 | Paranaíba | Medium (15%, 16%, 17%, 18%, 19%, 20%, 21%, 22%, 23%, 24%, and 25%) |
| 7 | Goiás | Small (10%, 11%, 12%, 13%, 14%, and 15%) |
| 8 | Rio de Janeiro | Large (40%, 41%, 42%, 43%, 44%, and 45%) |

The dimensions of the examined modular construction case study are proposed as 3 m × 3 m, with a total floor area of 9 m$^2$, and height of 3 m, as presented in Figure 2a. It could be used for administrative, residential, retail, or commercial services [36]. In this work, the case study is a building intended for residential purposes that users could occupy seven days a week and twenty-four hours per day. The facades of the examined building are within the size of 3 m × 3 m. The work will focus on locating the openings on one of the facades while the other facades could be stacked with other similar units (i.e., horizontal or vertical) to form the modeling of a modular construction building, as illustrated in Figure 2b. According to Ferrer [63], such a proposal could meet the need to arrange the modules in corridor form or around a central core. This work considers that the ventilation openings are positioned toward the magnetic north, using an open window that could ensure the use of 100% of the openings. At this level of analysis, the positioning of ventilation openings on the north side of construction buildings in these cities is indicated to best enhance the annual ventilation and natural daylight conditions [14]; however, the proposed framework of this work facilitates examining any other geographical positioning of openings.

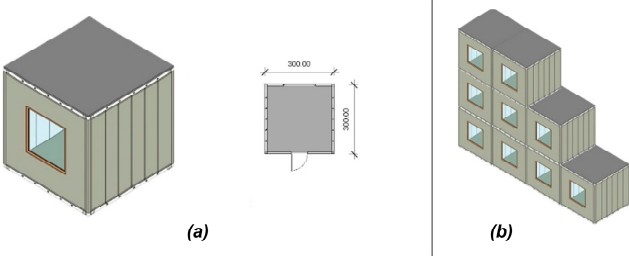

**Figure 2.** 2D and 3D modeling of the case study unit.

This work will, exceptionally, consider the total floor area of the built environment of the case study as 3 m × 3 m = 9 m², ignoring the thickness of the exterior walls; hence, the opening dimensions of the selected unit are varied based on the ventilation opening values presented in Table 1. Accordingly, the dimensions of the ventilation openings applied in this work will consider three different heights: 1.00 m for small ventilation openings; 1.20 m for medium ventilation openings; and 1.50 m for large ventilation openings. Table 2 illustrates that the eleven values of the medium ventilation openings (15%, 16%, 17%, 18%, 19%, 20%, 21%, 22%, 23%, 24%, and 25%) examined in the 6 cities of Curitiba, Rio Negro, São Paulo, Brasília, Campos, and Paranaíba, which necessitates simulating a total of 66 sequences (11 sequences for each of the presented six cities). This means that the medium ventilation openings of 15% require an opening size of 9 m² × 15% = 1.35 m². Taking into consideration that the proposed height for the medium ventilation openings is 1.20 m, the width of the ventilation opening for this sequence would be 1.35 m²/1.20 m = 1.125 m. Similarly, medium ventilation openings of 16% require an opening size of 1.20 m × 1.20 m; medium ventilation openings of 17% require an opening size of 1.275 m × 1.20 m; medium ventilation openings of 18% require an opening size of 1.35 m × 1.20 m; medium ventilation openings of 19% require an opening size of 1.425 m × 1.20 m; medium ventilation openings of 20% require an opening size of 1.50 m × 1.20 m; medium ventilation openings of 21% require an opening size of 1.575 m × 1.20 m; medium ventilation openings of 22% require an opening size of 1.65 m × 1.20 m; medium ventilation openings of 23% require an opening size of 1.725 m × 1.20 m; medium ventilation openings of 24% require an opening size of 1.80 m × 1.20 m; and medium ventilation openings of 25% require an opening size of 1.875 m × 1.20 m.

Table 2 highlights that the six values of the small ventilation openings (10%, 11%, 12%, 13%, 14%, and 15%) examined in the city of Goiás necessitate simulating a total of six sequences. This means that the small ventilation openings of 10% require an opening size of 9 m² × 10% = 0.90 m². Taking into consideration that the proposed height for the small ventilation openings is 1.00 m, the width of the ventilation opening for this sequence would be 0.90 m²/1.00 m = 0.90 m. Similarly, small ventilation openings of 11% require an opening size of 0.99 m × 1.00 m; small ventilation openings of 12% require an opening size of 1.08 m × 1.00 m; small ventilation openings of 13% require an opening size of 1.17 m × 1.00 m; small ventilation openings of 14% require an opening size of 1.26 m × 1.00 m; and small ventilation openings of 15% require an opening size of 1.35 m × 1.00 m.

Furthermore, Table 2 shows that the six values of the large ventilation openings (40%, 41%, 42%, 43%, 44%, and 45%) examined in the city of Rio de Janeiro necessitate simulating a total of six sequences. This means that the large ventilation openings of 40) require an opening size of 9 m² × 40% = 3.60 m². Taking into consideration that the proposed height for the large ventilation openings is 1.50 m, the width of the ventilation opening for this sequence would be 3.60 m²/1.50 m = 2.40 m. Similarly, large ventilation openings of 41% require an opening size of 2.46 m × 1.50 m; large ventilation openings of 42% require an opening size of 2.52 m × 1.50 m; large ventilation openings of 43% require an opening size of 2.58 m × 1.50 m; large ventilation openings of 44% require an opening

size of 2.64 m × 1.50 m; and large ventilation openings of 45% require an opening size of 2.70 m × 1.50 m.

The simulated modeling of the case study in this work is performed using Autodesk Revit as a BIM software, considering the use of metal frame construction for exterior walls and roof with a 3.0 degree pitch, single glazed sliding windows, and vinyl floor covering. The next step is to use Autodesk Revit to simulate the ventilation openings presented in Table 1 based on the referenced cities and dimensions illustrated in Table 2.

Accordingly, a total of 78 sequences were required to be modeled to simulate the case study. The final modeling of each sequence in Autodesk Revit should be exported as a gbXML document to be importable into Autodesk Green Building Studio, used in this study to evaluate the energy efficiency of construction projects. This step could facilitate comparing the variations in energy consumption of the same building in each city based on several ventilation openings, as well as comparing the best values over the eight selected cities for this work.

**Table 2.** Required sequences to be modeled in the examined cities.

| Sequence | Size of the Ventilation Opening in the Façade (m$^2$) | Dimension of the Opening (m) |
|:---:|:---:|:---:|
| Required sequences to be modeled in the cities of Curitiba, Rio Negro, São Paulo, Brasília, Campos, and Paranaíba | | |
| 1 | Medium ventilation openings of (15%) | 1.125 × 1.20 |
| 2 | Medium ventilation openings of (16%) | 1.20 × 1.20 |
| 3 | Medium ventilation openings of (17%) | 1.275 × 1.20 |
| 4 | Medium ventilation openings of (18%) | 1.35 × 1.20 |
| 5 | Medium ventilation openings of (19%) | 1.425 × 1.20 |
| 6 | Medium ventilation openings of (20%) | 1.50 × 1.20 |
| 7 | Medium ventilation openings of (21%) | 1.575 × 1.20 |
| 8 | Medium ventilation openings of (22%) | 1.65 × 1.20 |
| 9 | Medium ventilation openings of (23%) | 1.725 × 1.20 |
| 10 | Medium ventilation openings of (24%) | 1.80 × 1.20 |
| 11 | Medium ventilation openings of (25%) | 1.875 × 1.20 |
| Required sequences to be modeled in the city of Goiás | | |
| 1 | Small ventilation openings of (10%) | 0.90 × 1.00 |
| 2 | Small ventilation openings of (11%) | 0.99 × 1.00 |
| 3 | Small ventilation openings of (12%) | 1.08 × 1.00 |
| 4 | Small ventilation openings of (13%) | 1.17 × 1.00 |
| 5 | Small ventilation openings of (14%) | 1.26 × 1.00 |
| 6 | Small ventilation openings of (15%) | 1.35 × 1.00 |
| Required sequences to be modeled in the city of Rio de Janeiro | | |
| 1 | Large ventilation openings of (40%) | 2.40 × 1.50 |
| 2 | Large ventilation openings of (41%) | 2.46 × 1.50 |
| 3 | Large ventilation openings of (42%) | 2.52 × 1.50 |
| 4 | Large ventilation openings of (43%) | 2.58 × 1.50 |
| 5 | Large ventilation openings of (44%) | 2.64 × 1.50 |
| 6 | Large ventilation openings of (45%) | 2.70 × 1.50 |

## 5. Results and Discussion

The 78 sequences were modeled in Autodesk Revit software and simulated in Autodesk Green Building Studio. Each simulation resulted in an individual evaluation of

annual electricity consumption, as shown in Table 3. The cities of Curitiba, Rio Negro, São Paulo, Brasília, Campos, and Paranaíba were simulated in eleven sequences, while Goiás and Rio de Janeiro were replicated by six sequences based on Table 2.

**Table 3.** Annual electricity consumption.

| Analysis of Annual Electricity Consumption (kWh) | | | | | | | |
|---|---|---|---|---|---|---|---|
| Sequences | Curitiba | Rio Negro | São Paulo | Brasília | Campos | Paranaíba | Goiás | Rio de Janeiro |
| 1 | 3648 | 3033 | 4038 | 3917 | 4997 | 4913 | 5151 | 6111 |
| 2 | 3663 | 3048 | 4055 | 3931 | 5021 | 4933 | 5168 | 6139 |
| 3 | 3679 | 3065 | 4074 | 3946 | 5047 | 4957 | 5185 | 6169 |
| 4 | 3693 | 3080 | 4091 | 3960 | 5071 | 4977 | 5201 | 6199 |
| 5 | 3710 | 3096 | 4111 | 3976 | 5098 | 5000 | 5218 | 6229 |
| 6 | 3724 | 3109 | 4128 | 3990 | 5121 | 5020 | 5234 | 6259 |
| 7 | 3741 | 3124 | 4147 | 4005 | 5147 | 5043 | - | - |
| 8 | 3756 | 3139 | 4163 | 4019 | 5170 | 5063 | - | - |
| 9 | 3773 | 3156 | 4183 | 4035 | 5196 | 5085 | - | - |
| 10 | 3788 | 3169 | 4200 | 4048 | 5219 | 5105 | - | - |
| 11 | 3804 | 3186 | 4220 | 4064 | 5246 | 5128 | - | - |

Analyzing the annual electricity consumption in the selected cities necessitates assessing the end-use purposes for the energy consumed in construction projects, which can be obtained directly in Autodesk Green Building Studio for each simulation of the highlighted sequences. At this level of analysis, Table 4 evaluates the annual electricity consumption for lighting purposes as a percentage of the total annual electricity consumption of the case study building simulated in each of the examined cities based on the different ventilation openings proposed in NBR 15220.

**Table 4.** Annual electricity end use for lighting.

| Annual Electricity End Use for Lighting (%) | | | | | | | |
|---|---|---|---|---|---|---|---|
| Sequences | Curitiba | Rio Negro | São Paulo | Brasília | Campos | Paranaíba | Goiás | Rio de Janeiro |
| 1 | 62.5 | 64.3 | 73.2 | 72.4 | 78.4 | 78 | 79 | 82.3 |
| 2 | 62.6 | 64.5 | 73.3 | 72.5 | 78.5 | 78.1 | 79.1 | 82.4 |
| 3 | 62.8 | 64.7 | 73.5 | 72.6 | 78.6 | 78.2 | 79.1 | 82.5 |
| 4 | 63 | 64.9 | 73.6 | 72.7 | 78.7 | 78.3 | 79.2 | 82.6 |
| 5 | 63.1 | 65.1 | 73.7 | 72.8 | 78.8 | 78.4 | 79.3 | 82.6 |
| 6 | 63.3 | 65.2 | 73.8 | 72.9 | 78.9 | 78.5 | 79.3 | 82.7 |
| 7 | 63.4 | 65.4 | 73.9 | 73 | 79 | 78.5 | - | - |
| 8 | 63.6 | 65.5 | 74 | 73.1 | 79.1 | 78.6 | - | - |
| 9 | 63.7 | 65.7 | 74.1 | 73.2 | 79.2 | 78.7 | - | - |
| 10 | 63.9 | 65.9 | 74.2 | 73.3 | 79.3 | 78.8 | - | - |
| 11 | 64 | 66 | 74.4 | 73.4 | 79.4 | 78.9 | - | - |

Similarly, Table 5 evaluates the annual electricity consumption for heating ventilation and air conditioning (HVAC) purposes as a percentage of the annual electricity consumption of the case study building simulated in each of the examined cities based on the different ventilation openings proposed in NBR 15220. At this level of the study, the output

results show that using a small, medium, and large ventilation opening could significantly affect the annual energy consumption of the case study building in all cities.

**Table 5.** Annual electricity end use for HVAC.

| | | | | Annual Electricity End Use for HVAC (%) | | | | |
|---|---|---|---|---|---|---|---|---|
| Sequences | Curitiba | Rio Negro | São Paulo | Brasília | Campos | Paranaíba | Goiás | Rio de Janeiro |
| 1 | 16.3 | 17.8 | 13.4 | 13.8 | 10.8 | 11 | 10.5 | 8.9 |
| 2 | 16.2 | 17.7 | 13.3 | 13.8 | 10.8 | 11 | 10.5 | 8.8 |
| 3 | 16.2 | 17.6 | 13.3 | 13.7 | 10.7 | 10.9 | 10.4 | 8.8 |
| 4 | 16.1 | 17.6 | 13.2 | 13.7 | 10.7 | 10.9 | 10.4 | 8.7 |
| 5 | 16 | 17.5 | 13.2 | 13.6 | 10.6 | 10.8 | 10.4 | 8.7 |
| 6 | 16 | 17.4 | 13.1 | 13.6 | 10.6 | 10.8 | 10.3 | 8.6 |
| 7 | 15.9 | 17.3 | 13 | 13.5 | 10.5 | 10.7 | - | - |
| 8 | 15.8 | 17.2 | 13 | 13.5 | 10.5 | 10.7 | - | - |
| 9 | 15.8 | 17.1 | 12.9 | 13.4 | 10.4 | 10.6 | - | - |
| 10 | 15.7 | 17.1 | 12.9 | 13.4 | 10.4 | 10.6 | - | - |
| 11 | 15.6 | 17 | 12.8 | 13.3 | 10.3 | 10.5 | - | - |

In the city of Curitiba, the annual energy consumption varies between 3648 kWh/year, for a medium ventilation opening of 15%, and 3804 kWh/year, for a medium ventilation opening of 25%, as presented in Figure 3. Such a result illustrates the possibility of enhancing the annual energy consumption by around 4.27% when using the smaller size of the indicated openings in the façade (15%) rather than the larger sizes (between 16% and 25%) that are allowed in ABNT NBR 15220 for the medium ventilation openings. Figure 3 demonstrates that the annual electricity end use for HVAC systems represents around 62.5% and 64% of annual energy consumption for ventilation openings of 15% and 25%, respectively. Hence, achieving energy efficiency of 1.5% in such types of buildings based on the use of the HVAC systems requires using the smallest sizes of the indicated openings (15%). However, Figure 3 shows that the annual electricity end use for lighting represents around 16.3% and 15.6% of annual energy consumption for ventilation openings of 15% and 25%, respectively. It can be noted that increasing the size of ventilation openings in such types of buildings could facilitate the access of a high quantity of natural daylight to the inside environment, which could reduce the annual end use of electricity for lighting purposes by around 0.7%.

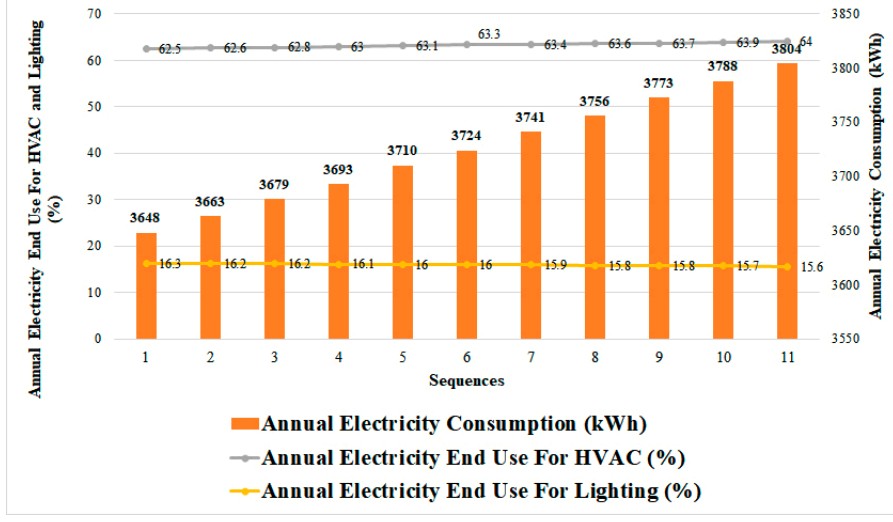

**Figure 3.** Annual electricity end-use consumption in Curitiba.

In the city of Rio Negro, the annual energy consumption varies between 3033 kWh/year, for a medium ventilation opening of 15%, and 3186 kWh/year, for a medium ventilation opening of 25%, as presented in Figure 4. These results highlight the possibility of improving the annual energy consumption by around 5.04% when using the ventilation openings in the façade of 15% rather than the larger sizes (between 16% and 25%) that are allowed in ABNT NBR 15220 for the medium ventilation openings. Figure 4 demonstrates that the annual electricity end use for HVAC systems is around 64.3% and 66% of annual energy consumption for ventilation openings of 15% and 25%, respectively. Hence, achieving energy efficiency of around 1.7% in such types of buildings based on the use of HVAC systems requires using the smallest sizes of the indicated openings (15%). However, Figure 4 shows that the annual electricity end use for lighting purposes represents around 16.3% and 15.6% of annual energy consumption for ventilation openings of 15% and 25%, respectively. Therefore, it can be seen that increasing the size of openings in such types of buildings could facilitate access to a high quantity of natural daylight, which could reduce the annual end use of electricity for lighting purposes by around 0.8%.

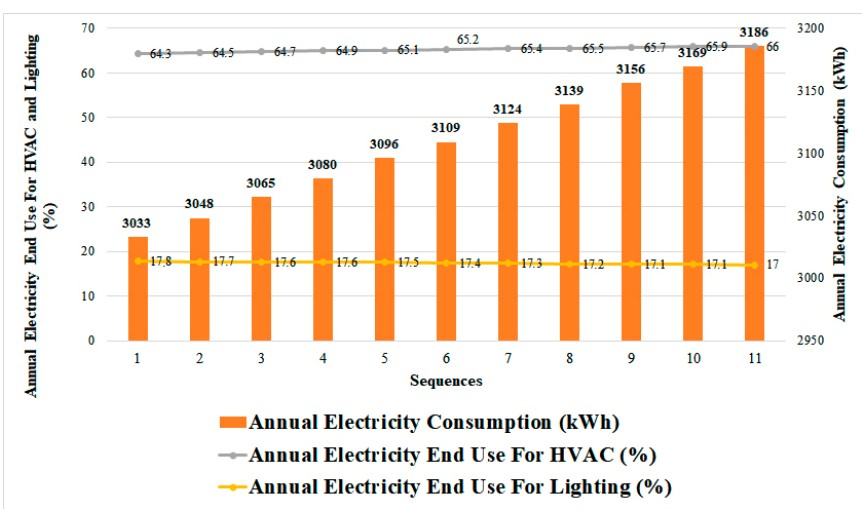

**Figure 4.** Annual electricity end-use consumption in Rio Negro.

In the city of São Paulo, the annual energy consumption varies between 4038 kWh/year, for a medium ventilation opening of 15%, and 4220 kWh/year, for a medium ventilation opening of 25%, as presented in Figure 5. These results highlight the possibility of improving the annual energy consumption by around 4.50% when using the lower sizes of ventilation openings in the façade of 15% rather than the larger sizes (between 16% and 25%) that are presented in ABNT NBR 15220 for the medium ventilation openings. Figure 5 demonstrates that the annual electricity end use for HVAC systems is around 73.2% and 74.4% of annual energy consumption for ventilation openings of 15% and 25%, respectively. Therefore, achieving energy efficiency of around 1.2% in such types of buildings based on using HVAC systems requires using the smallest sizes of indicated openings (15%). However, Figure 5 shows that the annual electricity end use for lighting purposes represents around 13.4% and 12.8% of annual energy consumption for ventilation openings of 15% and 25%, respectively. Therefore, it can be seen that increasing the size of openings in such types of buildings could facilitate access to a high quantity of natural daylight, which could reduce the annual end use of electricity for lighting purposes by around 0.6%.

In the city of Brasília, the annual energy consumption varies between 3917 kWh/year, for a medium ventilation opening of 15%, and 4064 kWh/year, for a medium ventilation opening of 25%, as presented in Figure 6. Such a result expresses the possibility of enhancing the annual energy consumption by around 3.75% when using the ventilation openings in the façade of 15% rather than other sizes (between 16% and 25%) that are indicated in ABNT NBR 15220 for the medium ventilation openings. Figure 6 demonstrates that the

annual electricity end use for HVAC systems is around 72.4% and 73.4% of annual energy consumption for ventilation openings of 15% and 25%, respectively. Hence, achieving energy efficiency of 1.0% in such types of buildings based on using HVAC systems requires using the smallest sizes of indicated openings (15%). However, Figure 6 shows that the annual electricity end use for lighting purposes represents around 13.8% and 13.3% of annual energy consumption for ventilation openings of 15% and 25%, respectively. Therefore, it can be seen that increasing the size of openings and using the ventilation openings of 25%, in such types of buildings could facilitate access to a high quantity of natural daylight, which could reduce the annual end use of electricity for lighting purposes by around 0.5%.

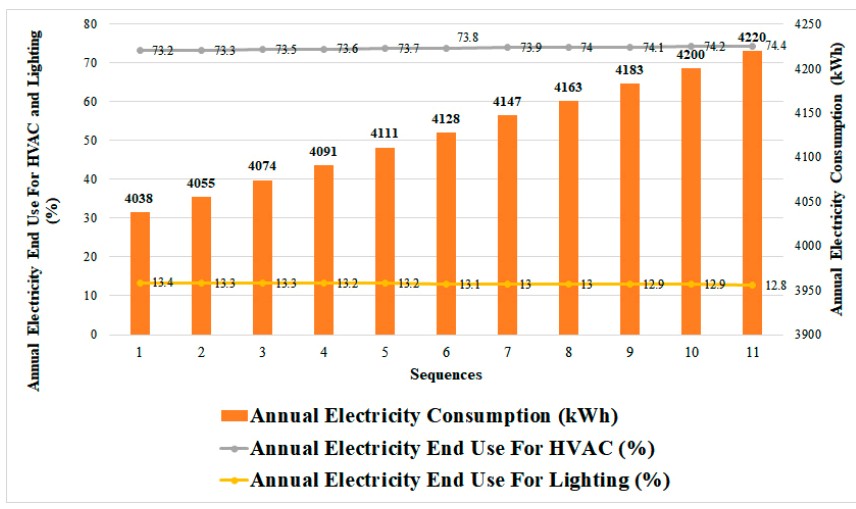

**Figure 5.** Annual electricity end-use consumption in São Paulo.

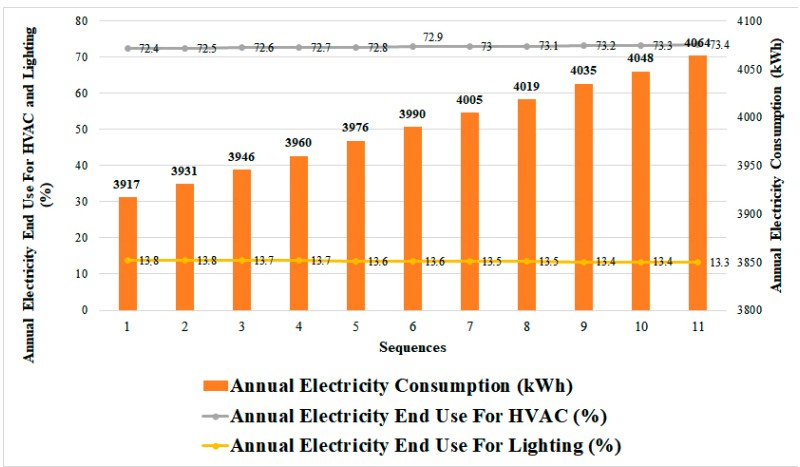

**Figure 6.** Annual electricity end-use consumption in Brasília.

In the city of Campos, the annual energy consumption varies between 4997 kWh/year, for a medium ventilation opening of 15%, and 5246 kWh/year, for a medium ventilation opening of 25%, as presented in Figure 7. Such a result expresses the possibility of enhancing the annual energy consumption by around 4.98% when using the indicated ventilation openings in the façade of 15% rather than the larger sizes (between 16% and 25%) that are highlighted in ABNT NBR 15220 for the medium ventilation openings. Figure 7 demonstrates that the annual electricity end use for HVAC systems varies between 78.4% and 79.4% of annual energy consumption for ventilation openings of 15% and 25%, respectively. Achieving an energy efficiency of around 1.0% in such types of buildings based on using the HVAC systems requires using the smallest sizes of the indicated openings (15%). However,

Figure 7 shows that the annual electricity end use for lighting purposes represents around 10.8% and 10.4% of annual energy consumption for ventilation openings of 15% and 25%, respectively. Hence, it can be seen that increasing the size of openings in such types of buildings could facilitate access to a high quantity of natural daylight, reducing the annual end use of electricity for lighting purposes by around 0.4%.

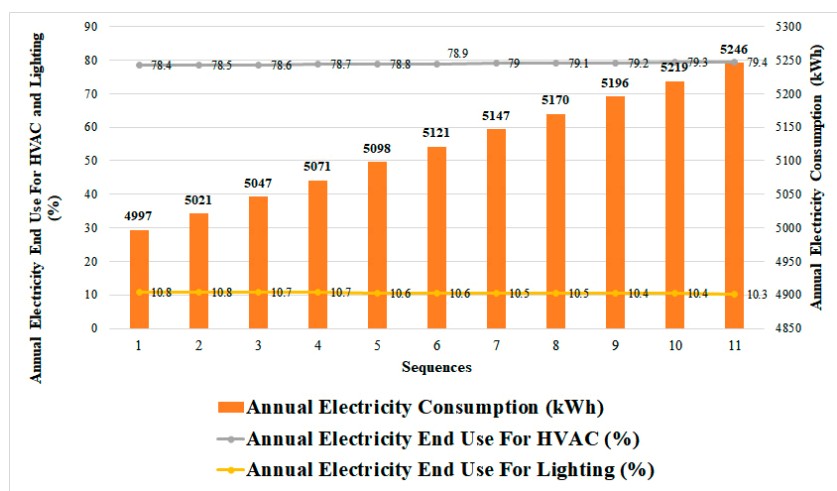

**Figure 7.** Annual electricity end-use consumption in Campos.

In the city of Paranaíba, the annual energy consumption varies between 4913 kWh/year, for a medium ventilation opening of 15%, and 5128 kWh/year, for a medium ventilation opening of 25%, as presented in Figure 8. Such a result expresses the possibility of enhancing the annual energy consumption by around 4.37% when using the lower sizes of ventilation openings in the façade (15%) rather than using the larger sizes (between 16% and 25%) that are indicated in ABNT NBR 15220 for the medium ventilation openings. Figure 8 demonstrates that the annual electricity end use for HVAC systems varies between 78.0% and 78.9% of annual energy consumption for ventilation openings of 15% and 25%, respectively. Hence, achieving energy efficiency of around 0.9% in such types of buildings based on using HVAC systems requires using smaller sizes of the indicated openings (15%). Figure 8 shows that the annual electricity end use for lighting purposes represents around 11.0% and 10.5% of annual energy consumption for ventilation openings of 15% and 25%, respectively. Therefore, it can be seen that increasing the size of openings in such types of buildings could facilitate access to a high quantity of natural daylight, reducing the annual end use of electricity for lighting purposes by around 0.5%.

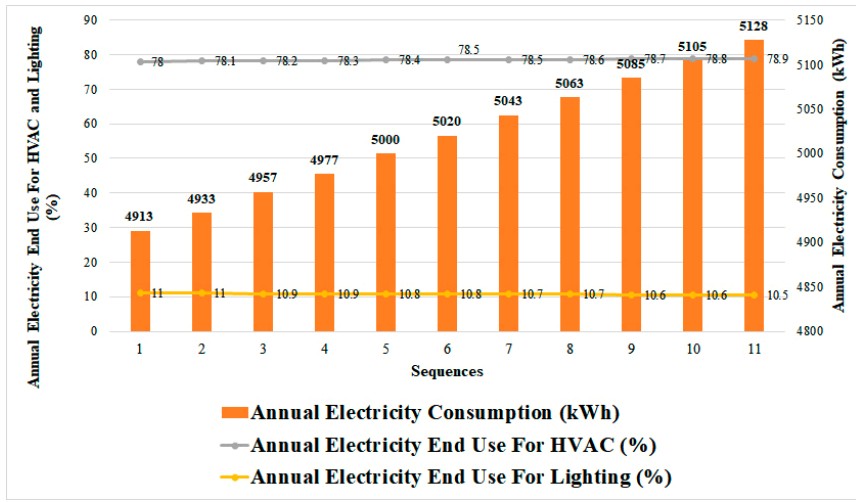

**Figure 8.** Annual electricity end-use consumption in Paranaíba.

In the city of Goiás, the annual energy consumption varies between 5151 kWh/year, for a small ventilation opening of 10%, and 5234 kWh/year, for a small ventilation opening of 15%, as presented in Figure 9. These results illustrate the possibility of enhancing the annual energy consumption by around 1.61% when using the lower indicated ventilation openings in the façade (10%) rather than other sizes (between 11% and 15%) that are allowed and presented in ABNT NBR 15220 for the small ventilation openings. Figure 9 demonstrates that the annual electricity end use for HVAC systems varies between 79.0% and 79.3% of annual energy consumption for ventilation openings of 10% and 15%, respectively. Hence, achieving energy efficiency of 0.3% in such types of constructions when using the HVAC systems requires using the smallest sizes of the indicated openings (10%). Figure 9 shows that the annual electricity end use for lighting purposes represents around 10.5% and 10.3% of annual energy consumption for ventilation openings of 10% and 15%, respectively. Therefore, it can be seen that increasing the size of openings in such types of buildings could facilitate access to a high quantity of natural daylight, reducing the annual end use of electricity for lighting purposes by around 0.2%.

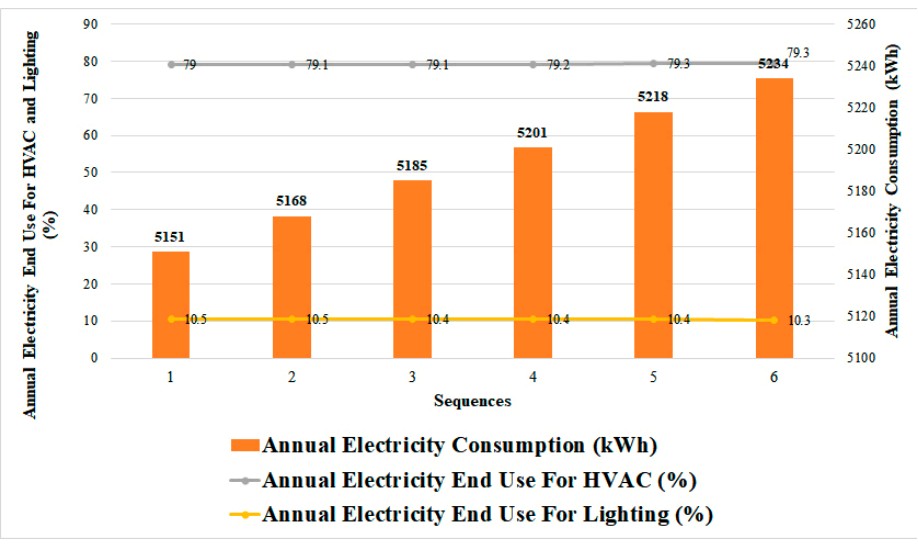

**Figure 9.** Annual electricity end-use consumption in Goiás.

In the city of Rio de Janeiro, the annual energy consumption varies between 6111 kWh/year, for a large ventilation opening of 40%, and 6259 kWh/year, for a large ventilation opening of 45%, as shown in Figure 10. These results show the possibility of improving the annual energy consumption by around 2.42% when using the smaller sizes of ventilation openings (40%) in the façade rather than any other larger sizes indicated in ABNT NBR 15220 for the large ventilation openings. Figure 10 demonstrates that the annual electricity end use for HVAC systems varies between 82.3% and 82.7% of annual energy consumption for ventilation openings of 40% and 45%, respectively. Therefore, when using the HVAC systems, achieving energy efficiency of around 0.4% in such types of buildings requires using the smallest sizes of the indicated openings (40%). Figure 10 shows that the annual electricity end use for lighting purposes represents around 8.9% and 8.6% of annual energy consumption for ventilation openings of 40% and 45%, respectively. Hence, it can be noted that increasing the size of openings in such types of buildings could facilitate access to a high quantity of natural daylight, which could reduce the annual end use of electricity for lighting purposes by around 0.3%.

To sum up, it can be noted that based on the ventilation openings given in ABNT NBR 15220, using the smallest sizes instead of the indicated medium ones could improve the annual energy efficiency between 3.75% and 5.04% in the examined cities. This value could be reduced to 1.61% in an examined city where the use of small ventilation openings in building construction is indicated, while in a city where the use of large ventilation

openings is indicated could achieve an energy efficiency of 2.42% when using the smallest sizes of openings. Hence, it is important to consider the size of ventilation openings in modular construction projects to reduce the energy consumption in such types of buildings. At the same time, the major role of other construction components that make up the building's envelope (i.e., external wall, roof, floor, and windows) in producing energy-efficient buildings should be taken into account.

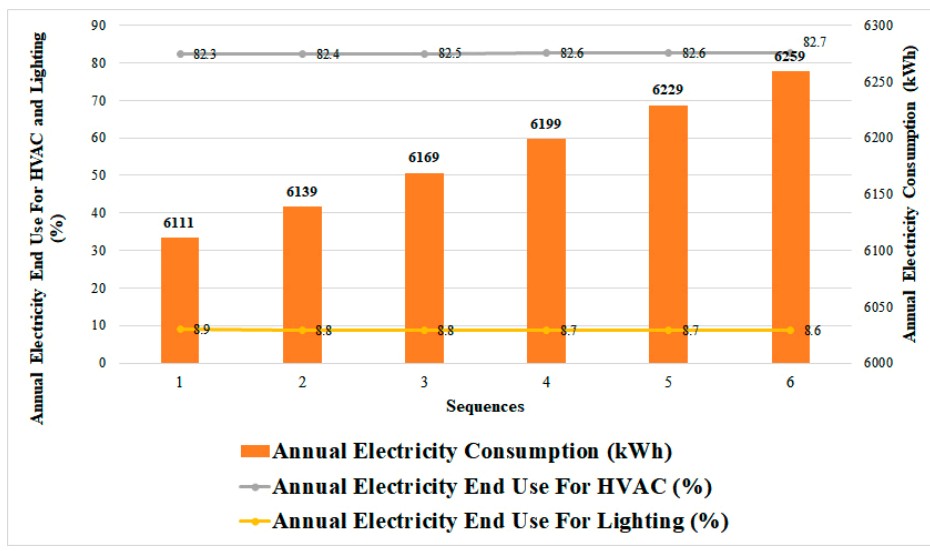

**Figure 10.** Annual electricity end-use consumption in Rio de Janeiro.

Figure 11 illustrates the electricity consumption analysis of the modular construction building used in the case study in the examined cities based on the lower values of the ventilation openings indicated in ABNT NBR 15220. It can be seen that the energy consumption of the same building model would vary between the examined cities. For example, the modular construction building using the smallest ventilation openings would consume up to 6111 kWh/year in Rio de Janeiro, which is located in the 8th BioZ and where the use of large ventilation openings is indicated. However, these values would be reduced to 3033 kWh/year in the city of Rio Negro, which is located in the 2nd BioZ and where the use of medium ventilation openings is indicated. Generally, Campos (4997 kWh/year), Paranaíba (4913 kWh/year), and Goiás (5151 kWh/year) are identified as having high energy consumption. These cities are located in the 5th, 6th, and 7th BioZ, respectively, and different sizes of ventilation openings are recommended. On the other side, the energy consumption of the case study building in the cities of Curitiba (3648 kWh/year), São Paulo (4038 kWh/year), and Brasília (3917 kWh/year) illustrates some better results. These cities are located in the 1st, 3rd, and 4th BioZ, respectively, and the use of medium ventilation openings is indicated. This means that the modeling of the same modular construction unit, using the smallest ventilation openings indicated by ABNT NBR 15220 in the city of Rio Negro (2nd BioZ) could consume around 50% of energy compared to constructing the same building with the smallest ventilation openings in the city of Rio de Janeiro (8th BioZ). Similarly, modeling the construction unit with the smallest size of ventilation openings indicated in ABNT NBR 15220 in Curitiba (1st BioZ), São Paulo (3rd BioZ), Brasília (4th BioZ), Campos (5th BioZ), Paranaíba (6th BioZ), and Goiás (7th BioZ) could consume less energy of around 40%, 34%, 36%, 18%, 20%, and 16%, respectively, compared to constructing the same building in the city of Rio de Janeiro (8th BioZ). Hence, it is highly important to consider the BioZ of the location of the construction project as a major parameter that could influence the energy consumption in these projects, considering the goal of energy-efficient buildings.

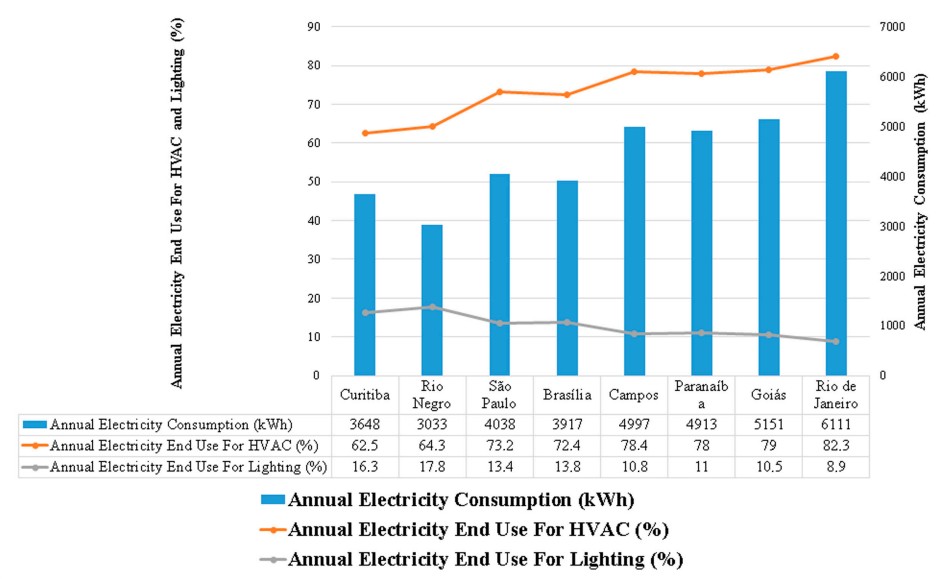

**Figure 11.** Electricity consumption analysis of the case study in the examined cities.

## 6. Conclusions

The aim of this study is to facilitate the decision-making process in terms of proposing the best ventilation opening dimensions for sustainable energy use and management in buildings. Hence, a novel framework is presented to evaluate the impact and propose the best dimensions of ventilation openings for metal frame modular construction in several Brazilian bioclimatic zones based on ABNT NBR 15220, using BIM.

The proposed framework uses Autodesk Revit as a BIM tool to simulate the modular construction projects and the relevant dimension of the ventilation opening as proposed in the Brazilian standard ABNT NBR 15220, in several Brazilian cities that are located in different bioclimatic zones: Curitiba in the first zone; Rio Negro in the second zone; São Paulo in the third zone; Brasilia in the fourth zone; Campos in the fifth zone; Paranaiba in the sixth zone; Goiás in the seventh zone; and Rio de Janeiro in the eighth zone. The Autodesk Green Building Studio is applied as another BIM tool that could evaluate the energy consumption in the simulated case study building according to each bioclimatic zone in each of the examined cities. The major conclusions of this work are as follows:

- The energy consumption of the same building varies between the examined cities based on their bioclimatic classification zone.
- Using the lowest values of ventilation openings as indicated in ABNT NBR 15220 for the different bioclimatic zones would increase energy efficiency in buildings.
- The bioclimatic zone of the location of the construction project could influence energy consumption, affecting the energy efficiency of buildings.

The results of this work could help practitioners and professionals working on modular construction projects in Brazil to design the best dimensions of ventilation openings based on each BioZ. At this level of analysis, it is important to highlight that the framework proposed herein could be applied to any other construction method (i.e., conventional construction, modern construction, etc.) located in several regions or cities within different bioclimatic zones of Brazil. However, using the Brazilian standard ABNT NBR 15220 produced two major limitations of this study, as follows:

- Dedicating the application of the proposed framework to examining the energy efficiency of ventilation openings in Brazilian cities only.
- Taking into consideration only the minimum and maximum values of ventilation openings, as indicated in ABNT NBR 15220 for the small, medium, and large openings.

Hence, a recommendation for future works could be to examine the application of the same framework in other countries, taking into consideration the related and local

standards for ventilation openings in the same country. In addition, using different window-to-wall ratios within higher or lower values of ventilation openings could be considered as another recommendation for future work aimed at improving the energy efficiency of such types of buildings. Furthermore, examining the impact of other building components together with the size of ventilation openings on the energy consumption in buildings could be another recommendation at this level of the study.

**Author Contributions:** Conceptualization, M.K.N., L.O.C.D.A. and A.H.; methodology, M.K.N. and A.H.; software, M.K.N.; validation, M.K.N., L.O.C.D.A., O.O., M.K., C.A.P.S., D.B., K.V.F. and A.H.; formal analysis, M.K.N., L.O.C.D.A., O.O., M.K., D.B., K.V.F., C.A.P.S. and A.H.; investigation, M.K.N., L.O.C.D.A., D.B., C.A.P.S. and A.H.; data curation, M.K.N., L.O.C.D.A., O.O., M.K., C.A.P.S., D.B., K.V.F. and A.H.; writing—original draft preparation, M.K.N.; writing—review and editing, M.K.N., D.B., C.A.P.S. and A.H.; visualization, M.K.N., L.O.C.D.A., C.A.P.S. and A.H.; supervision, M.K.N., L.O.C.D.A., D.B., C.A.P.S. and A.H. All authors have read and agreed to the published version of the manuscript.

**Funding:** This study was funded by the National Council for Scientific and Technological Development—CNPq—Brazil (grants 314085/2020-3, 304726/2021-4) and FAPERJ (Fundação de Amparo à Pesquisa do Estado do Rio de Janeiro) grant 2019-E-26/202.568/2019 (245653).

**Institutional Review Board Statement:** Not applicable.

**Informed Consent Statement:** Not applicable.

**Data Availability Statement:** Not applicable.

**Acknowledgments:** The authors would like to thank the National Council for Scientific and Technological Development—CNPq—Brazil for supporting the research reported in this paper. The authors also thank the editor and anonymous reviewers for their comments and suggestions.

**Conflicts of Interest:** The authors declare no conflict of interest.

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
