# Peer review of "Influence of Ventilation Openings on the Energy Efficiency of Metal Frame Modular Constructions in Brazil Using BIM"

_2673-4117, doi:10.3390/eng4020093_

Round 1
Reviewer 1 Report
Thank you for the opportunity to review this paper. There are some suggestions for improvement:
1. Introduction - please clearly state the knowledge gap, motivation of conducting this study, the aim and objectives.
2. Conclusions - please discuss the implications of the findings beyond Brazil. Can the framework be applied to other countries and other construction scenarios? Any limitations and further research?
Author Response
Responses to Reviewer 1:
Thank you for the opportunity to review this paper. There are some suggestions for improvement:
Response: The authors would like to thank the reviewer for his valuable comments that helped improve the quality of the manuscript.
- Introduction - please clearly state the knowledge gap, motivation of conducting this study, the aim and objectives.
Response: Thank you for your comment. The knowledge gap, motivation of study, aim and objectives have been addressed in the updated manuscript in the Introduction Section, as follows:
“In the literature, several studies have examined the important role of ventilation openings to increase energy efficiency in buildings [26], [27], [28], [29], [30], however, a gap of knowledge keens in proposing the best dimensions of the ventilation openings of construction projects toward more sustainable and energy-efficient buildings. Thus, this work aims to empower the decision-making process in terms of proposing the best ventilation opening dimensions toward sustainable energy use and management in buildings. A novel framework is presented herein to evaluate the impact and propose the best dimensions of ventilation openings of metal frame modular construction in several Brazilian bioclimatic zones (BioZ), using BIM. The motivation of this work is to help the practitioners and professionals in the modular construction project in Brazil to design the best dimensions of the ventilation opening based on each BioZ.”
- Conclusions - please discuss the implications of the findings beyond Brazil. Can the framework be applied to other countries and other construction scenarios? Any limitations and further research?
Response: Thank you for this valuable comment. The conclusions section has been revised to present the implications of this work and discuss the limitations and future recommendations as follows:
“The output results of this work could help the practitioners and professionals in the modular construction project in Brazil to design the best dimensions of the ventilation opening based on each BioZ. At this level of the analysis, it is important to highlight that the proposed framework herein could be applied to any other construction methods (i.e. conventional construction, modern construction, etc.) located in several regions or cities within different climate classifications and bioclimatic zones in Brazil. However, using the Brazilian standard ABNT NBR 15220 produced two major limitations for this study, as follows:
- Dedicating the application of the proposed framework to examine the energy efficiency of ventilation openings in Brazilian cities only.
- Taking into consideration the minimum and maximum values only of ventilation openings as indicated in ABNT NBR 15220 for the small, medium, and large openings.
Hence, the recommendations for future works could be in examining the application of the same framework in other countries, taking into consideration the related and local standards for ventilation openings in the same country. Besides, using different window-to-wall ratios within higher or lower values of ventilation openings could be considered as another recommendation for future work toward more energy efficiency in such types of buildings. Furthermore, examining the impact of other building components together with the ventilation-opening role on the energy consumption in buildings could be another recommendation at this level of the study.”

Reviewer 2 Report
The study proposes a framework to evaluate the impact of ventilation openings on the energy efficiency of metal frame modular construction in Brazil. The research shows that the energy consumption of the same building model varies based on the dimension of ventilation openings for each Bioclimatic Zone (BioZ) in Brazil.
Here are some recommendations for revision:
- Simplify the conclusion by presenting it in bullet point format.
- Since this study focuses on the energy efficiency of ventilation, it is important to include data on ventilation efficiency and associated energy consumption in the analysis.
Author Response
Responses to Reviewer 2:
The study proposes a framework to evaluate the impact of ventilation openings on the energy efficiency of metal frame modular construction in Brazil. The research shows that the energy consumption of the same building model varies based on the dimension of ventilation openings for each Bioclimatic Zone (BioZ) in Brazil.
Here are some recommendations for revision:
Response: The authors would like to thank the reviewer for the valuable comments that helped improve the quality of the manuscript.
- Simplify the conclusion by presenting it in bullet point format.
Response: Thank you for your comment. The conclusions section has been revised where the major results and the research limitations have been presented in bullet point format, as follows:
The major results:
“The major outputs of this work are as follows:
- The energy consumption of the same building varies between the examined cities based on their bioclimatic classification zone.
- Using the lowest values of ventilation openings as indicated in ABNT NBR 15220 for the different bioclimatic zones would increase energy efficiency in buildings.
- The bioclimatic zone of the location of the construction project could influence energy consumption toward energy-efficient buildings.”
The research limitations:
“The output results of this work could help the practitioners and professionals in the modular construction project in Brazil to design the best dimensions of the ventilation opening based on each BioZ. At this level of the analysis, it is important to highlight that the proposed framework herein could be applied to any other construction methods (i.e. conventional construction, modern construction, etc.) located in several regions or cities within different climate classifications and bioclimatic zones in Brazil. However, using the Brazilian standard ABNT NBR 15220 produced two major limitations for this study, as follows:
- Dedicating the application of the proposed framework to examine the energy efficiency of ventilation openings in Brazilian cities only.
- Taking into consideration the minimum and maximum values only of ventilation openings as indicated in ABNT NBR 15220 for the small, medium, and large openings.
Hence, the recommendations for future works could be in examining the application of the same framework in other countries, taking into consideration the related and local standards for ventilation openings in the same country. Besides, using different window-to-wall ratios within higher or lower values of ventilation openings could be considered as another recommendation for future work toward more energy efficiency in such types of buildings. Furthermore, examining the impact of other building components together with the ventilation-opening role on the energy consumption in buildings could be another recommendation at this level of the study.”
- Since this study focuses on the energy efficiency of ventilation, it is important to include data on ventilation efficiency and associated energy consumption in the analysis.
Response: Thank you for this valuable comment. The authors have conducted a deep analysis in the literature to include data on ventilation efficiency and associated energy consumption in buildings. The following text has been added to the subsection (2.2.):
“Ventilation openings have a major role in determining the energy efficiency in buildings and thermal comfort [47], [48]. Salcido et al. assessed the mixed ventilation systems for the integration of natural ventilation in office buildings, where, concerning windows, the wind behavior is fundamental for their design and controlling their opening period [49]. Lotfabadi and Hançer [50] developed a novel multi-objective optimization model of building openings to evaluate the visual comfort and energy efficiency of an office building in Cyprus. The authors highlighted that window-to-wall ratio and glazing type could play a major role in energy efficiency optimization in buildings. The output results illustrated that a window-to-wall ratio between 20% and 50% could increase the energy efficiency of the case study building in Cyprus. Tien et al [28] presented a vision-based deep learning framework for the detection and recognition of window operation toward effective management of ventilation heat losses in buildings. The authors confirmed that the proposed framework could increase energy efficiency in buildings through the detection of the dimension and types of windows. Matour et al [51] evaluated the impact of opening configurations on the ventilation performance of an integrated tall building with a double skin façade, taking into consideration the major four wind orientations; 0â—¦, 30â—¦, 60â—¦, and 90â—¦. The evaluation process was conducted based on three ventilation performance indicators that are induced air flow, wind speed ratio, and airflow distribution across the cavity length. Bauer et al. [52] analyzed the household practices and the construction components of the buildings and observed that the window's positioning could play a major role to assess the energy efficiency of construction projects, through the direct and indirect loss of heating energy in buildings. Dussault et al. analyzed different levels of window opacities and economy in the HVAC system, being beneficial in temperate climates, taking into account the window direction [9]. Fabi et al. identified and analyzed driving forces for the behavior of opening and closing windows, including physiological, psychological, social, physical, and contextual environment, and how these factors could influence ventilation and thermal control in residential buildings and offices [53].
Furthermore, BIM tools have been applied in the literature to assess the role of ventilation openings on energy efficiency in buildings. For instance, Kim et al [54] examined (65) different design scenarios based on the size, position, and orientation of windows in buildings, using Autodesk Green Building Studio as a BIM tool. The authors found that energy consumption in buildings would increase when using high values of window-to-wall ratio, and vice-versa. Gan et al [55] proposed a BIM-based framework to evaluate the natural ventilation impact on energy efficiency and thermal comfort in buildings. The authors highlighted that BIM tools could facilitate the evaluation process of natural ventilation to maintain indoor thermal comfort and reduce energy consumption in buildings. Zhang et al [56] prepared a BIM-based architectural analysis to optimize the building design and concept. The authors presented the important role of BIM to evaluate the effect of construction components in energy-efficient buildings, basically, the external walls and ventilation openings. Similarly, Maglad et al [57] elaborated a case study building to assess the impact of BIM tools on energy consumption evaluation in buildings, using Autodesk Insight 360 and Green Building Studio. The author confirmed that the BIM methodology could promote energy efficiency in construction projects.”

Reviewer 3 Report
My comments:
-Abstract must have problem statement.
-As seen from the ref list, many references are incomplete.
-Literature review is incomplete. The updated works on the topic are ignored.
-Table 2. Required sequences to be modeled in the examined cities 290 must be better explained.
-Conclusion is too long. Some parts must be moved to the discussion section.
-Research implications are not provided for the theory and method. In addition, limitations are poorly presented.
Author Response
Responses to Reviewer 3:
My comments:
Response: The authors would like to thank the reviewer for the valuable comments that helped improve the quality of the manuscript.
-Abstract must have problem statement.
Response: Thank you for your comment. The problem statement is being added in the updated manuscript at the beginning of the Abstract, as follows:
“Construction projects demands a higher amount of energy predominantly for heating, ventilation, and illuminance purposes. Modular construction comes to the limelight in recent years as a construction method that uses sustainable building materials and optimizes energy efficiency. Ventilation openings in buildings are designed to facilitate air circulation by naturally driven ventilation and could aid in reducing energy consumption in construction projects. However, a gap of knowledge is keen on proposing the best dimensions of ventilation openings in buildings. Hence, the aim of this work is to empower the decision-making process in terms of proposing the best ventilation opening dimension toward sustainable energy use and management in buildings. A novel framework presented herein to evaluate the impact and propose the best dimensions of ventilation openings of metal frame modular construction in Brazil, using Building Information Modeling.”
-As seen from the ref list, many references are incomplete.
Response: Thank you for this very important comment. The authors would like to apologize for such a technical issue. We have addressed this point in the updated manuscript where all of the used references have been revised using Mendeley.
-Literature review is incomplete. The updated works on the topic are ignored.
Response: Thank you for this valuable comment. The authors have conducted a deep analysis in the literature to include the required data and complete the analysis. The following text has been added to the subsection (2.2.):
“Ventilation openings have a major role in determining the energy efficiency in buildings and thermal comfort [47], [48]. Salcido et al. assessed the mixed ventilation systems for the integration of natural ventilation in office buildings, where, concerning windows, the wind behavior is fundamental for their design and controlling their opening period [49]. Lotfabadi and Hançer [50] developed a novel multi-objective optimization model of building openings to evaluate the visual comfort and energy efficiency of an office building in Cyprus. The authors highlighted that window-to-wall ratio and glazing type could play a major role in energy efficiency optimization in buildings. The output results illustrated that a window-to-wall ratio between 20% and 50% could increase the energy efficiency of the case study building in Cyprus. Tien et al [28] presented a vision-based deep learning framework for the detection and recognition of window operation toward effective management of ventilation heat losses in buildings. The authors confirmed that the proposed framework could increase energy efficiency in buildings through the detection of the dimension and types of windows. Matour et al [51] evaluated the impact of opening configurations on the ventilation performance of an integrated tall building with a double skin façade, taking into consideration the major four wind orientations; 0â—¦, 30â—¦, 60â—¦, and 90â—¦. The evaluation process was conducted based on three ventilation performance indicators that are induced air flow, wind speed ratio, and airflow distribution across the cavity length. Bauer et al. [52] analyzed the household practices and the construction components of the buildings and observed that the window's positioning could play a major role to assess the energy efficiency of construction projects, through the direct and indirect loss of heating energy in buildings. Dussault et al. analyzed different levels of window opacities and economy in the HVAC system, being beneficial in temperate climates, taking into account the window direction [9]. Fabi et al. identified and analyzed driving forces for the behavior of opening and closing windows, including physiological, psychological, social, physical, and contextual environment, and how these factors could influence ventilation and thermal control in residential buildings and offices [53].
Furthermore, BIM tools have been applied in the literature to assess the role of ventilation openings on energy efficiency in buildings. For instance, Kim et al [54] examined (65) different design scenarios based on the size, position, and orientation of windows in buildings, using Autodesk Green Building Studio as a BIM tool. The authors found that energy consumption in buildings would increase when using high values of window-to-wall ratio, and vice-versa. Gan et al [55] proposed a BIM-based framework to evaluate the natural ventilation impact on energy efficiency and thermal comfort in buildings. The authors highlighted that BIM tools could facilitate the evaluation process of natural ventilation to maintain indoor thermal comfort and reduce energy consumption in buildings. Zhang et al [56] prepared a BIM-based architectural analysis to optimize the building design and concept. The authors presented the important role of BIM to evaluate the effect of construction components in energy-efficient buildings, basically, the external walls and ventilation openings. Similarly, Maglad et al [57] elaborated a case study building to assess the impact of BIM tools on energy consumption evaluation in buildings, using Autodesk Insight 360 and Green Building Studio. The author confirmed that the BIM methodology could promote energy efficiency in construction projects.”
-Table 2. Required sequences to be modeled in the examined cities 290 must be better explained.
Response: Thank you for this very important comment. This Table is better explained now in the updated manuscript and the simulated sequences to be modeled in the examined cities are described as follows:
“Accordingly, the dimensions of the ventilation openings applied in this work will consider three different heights; (1,00m) for small ventilation openings; (1,20m) for medium ventilation openings; and (1,50m) for large ventilation openings. Table 2 illustrates that the eleven values of the medium ventilation openings (15%, 16%, 17%, 18%, 19%, 20%, 21%, 22%, 23%, 24%, and 25%) examined in the six cities; Curitiba, Rio Negro, São Paulo, Brasília, Campos, and Paranaíba, necessitates simulating a total of (66 sequences); eleven sequences for each of the presented six cities. This means that the medium ventilation openings of (15%) require an opening size of (9m2x15%=1,35m2). Taking into consideration that the proposed height for the medium ventilation openings is (1,20m), the width of the ventilation opening for this sequence would be (1,35m2/1,20m=1,125m). Similarly, medium ventilation openings of (16%) require an opening size of (1,20mx1,20m); medium ventilation openings of (17%) requires an opening size of (1,275mx1,20m); medium ventilation openings of (18%) requires an opening size of (1,35mx1,20m); medium ventilation openings of (19%) requires an opening size of (1,425mx1,20m); medium ventilation openings of (20%) requires an opening size of (1,50mx1,20m); medium ventilation openings of (21%) requires an opening size of (1,575mx1,20m); medium ventilation openings of (22%) requires an opening size of (1,65mx1,20m); medium ventilation openings of (23%) requires an opening size of (1,725mx1,20m); medium ventilation openings of (24%) requires an opening size of (1,80mx1,20m); and medium ventilation openings of (25%) requires an opening size of (1,875mx1,20m).
Table 2 highlights that the six values of the small ventilation openings (10%, 11%, 12%, 13%, 14%, and 15%) examined in the city of Goiás necessitate simulating a total of (6 sequences). This means that the small ventilation openings of (10%) requires an opening size of (9m2x10%=0,90m2). Taking into consideration that the proposed height for the small ventilation openings is (1,00m), the width of the ventilation opening for this sequence would be (0,90m2/1,00m=0,90m). Similarly, small ventilation openings of (11%) requires an opening size of (0,99mx1,00m); small ventilation openings of (12%) requires an opening size of (1,08mx1,00m); small ventilation openings of (13%) requires an opening size of (1,17mx1,00m); small ventilation openings of (14%) requires an opening size of (1,26mx1,00m); and small ventilation openings of (15%) requires an opening size of (1,35mx1,00m).
Furthermore, Table 2 presents that the six values of the large ventilation openings (40%, 41%, 42%, 43%, 44%, and 45%) examined in the city of Rio de Janeiro necessitate simulating a total of (6 sequences). This means that the large ventilation openings of (40%) requires an opening size of (9m2x40%=3,60m2). Taking into consideration that the proposed height for the large ventilation openings is (1,50m), the width of the ventilation opening for this sequence would be (3,60m2/1,50m=2,40m). Similarly, large ventilation openings of (41%) requires an opening size of (2,46mx1,50m); large ventilation openings of (42%) requires an opening size of (2,52mx1,50m); large ventilation openings of (43%) requires an opening size of (2,58mx1,50m); large ventilation openings of (44%) requires an opening size of (2,64mx1,50m); and large ventilation openings of (45%) requires an opening size of (2,70mx1,50m).”
-Conclusion is too long. Some parts must be moved to the discussion section.
Response: Thank you for your comment. The conclusions section has been revised according to your comment as follows:
“6. Conclusions
The aim of this study id to empower the decision-making process in terms of proposing the best ventilation opening dimensions toward sustainable energy use and management in buildings. Hence, a novel framework is presented to evaluate the impact and propose the best dimensions of ventilation openings of metal frame modular construction in several Brazilian bioclimatic zones based on ABNT NBR 15220., using BIM.
The proposed framework uses Autodesk Revit as a BIM tool to simulate the modular construction projects and the relevant dimension of the ventilation opening as proposed in the Brazilian standard ABNT NBR 15220in several Brazilian cities that are located in different bioclimatic zones; Curitiba in the first zone; Rio Negro in the second zone; São Paulo in the third zone; Brasilia in the fourth zone; Campos in the fifth zone; Paranaiba in the sixth zone; Goiás in the seventh zone; and Rio de Janeiro in the eighth zone. The Autodesk Green Building Studio is applied as another BIM tool that could evaluate the energy consumption in the simulated case study building according to each bioclimatic zone in each of the examined cities. The major outputs of this work are as follows:
- The energy consumption of the same building varies between the examined cities based on their bioclimatic classification zone.
- Using the lowest values of ventilation openings as indicated in ABNT NBR 15220 for the different bioclimatic zones would increase energy efficiency in buildings.
- The bioclimatic zone of the location of the construction project could influence energy consumption toward energy-efficient buildings.
The output results of this work could help the practitioners and professionals in the modular construction project in Brazil to design the best dimensions of the ventilation opening based on each BioZ. At this level of the analysis, it is important to highlight that the proposed framework herein could be applied to any other construction methods (i.e. conventional construction, modern construction, etc.) located in several regions or cities within different climate classifications and bioclimatic zones in Brazil. However, using the Brazilian standard ABNT NBR 15220 produced two major limitations for this study, as follows:
- Dedicating the application of the proposed framework to examine the energy efficiency of ventilation openings in Brazilian cities only.
- Taking into consideration the minimum and maximum values only of ventilation openings as indicated in ABNT NBR 15220 for the small, medium, and large openings.
Hence, the recommendations for future works could be in examining the application of the same framework in other countries, taking into consideration the related and local standards for ventilation openings in the same country. Besides, using different window-to-wall ratios within higher or lower values of ventilation openings could be considered as another recommendation for future work toward more energy efficiency in such types of buildings. Furthermore, examining the impact of other building components together with the ventilation-opening role on the energy consumption in buildings could be another recommendation at this level of the study.”
-Research implications are not provided for the theory and method. In addition, limitations are poorly presented.
Response: Thank you for this valuable comment. The research implications as well as the knowledge gap, aims and novelty have been added to the introduction section as follows:
“In the literature, several studies have examined the important role of ventilation openings to increase energy efficiency in buildings [26], [27], [28], [29], [30], however, a gap of knowledge keens in proposing the best dimensions of the ventilation openings of construction projects toward more sustainable and energy-efficient buildings. Thus, this work aims to empower the decision-making process in terms of proposing the best ventilation opening dimensions toward sustainable energy use and management in buildings. A novel framework is presented herein to evaluate the impact and propose the best dimensions of ventilation openings of metal frame modular construction in several Brazilian bioclimatic zones (BioZ), using BIM. The motivation of this work is to help the practitioners and professionals in the modular construction project in Brazil to design the best dimensions of the ventilation opening based on each BioZ.”
And to the methods section as follows:
“Achieving the aim of this work by empowering the decision-making process in terms of designing the ventilation opening dimensions toward sustainable energy use and management in buildings. This work proposes a framework that facilitates evaluating the impact and proposes the best dimensions of ventilation openings of metal frame modular construction in several Brazilian bioclimatic zones (BioZ) using BIM, as presented in Figure 1. Such a framework could help the practitioners and professionals in the modular construction project to design the best dimensions of the ventilation opening based on each BioZ towards increasing energy efficiency and sustainability.”
The research limitations have been revised in the updated manuscript as follows:
The output results of this work could help the practitioners and professionals in the modular construction project in Brazil to design the best dimensions of the ventilation opening based on each BioZ. At this level of the analysis, it is important to highlight that the proposed framework herein could be applied to any other construction methods (i.e. conventional construction, modern construction, etc.) located in several regions or cities within different climate classifications and bioclimatic zones in Brazil. However, using the Brazilian standard ABNT NBR 15220 produced two major limitations for this study, as follows:
- Dedicating the application of the proposed framework to examine the energy efficiency of ventilation openings in Brazilian cities only.
- Taking into consideration the minimum and maximum values only of ventilation openings as indicated in ABNT NBR 15220 for the small, medium, and large openings.
Hence, the recommendations for future works could be in examining the application of the same framework in other countries, taking into consideration the related and local standards for ventilation openings in the same country. Besides, using different window-to-wall ratios within higher or lower values of ventilation openings could be considered as another recommendation for future work toward more energy efficiency in such types of buildings. Furthermore, examining the impact of other building components together with the ventilation-opening role on the energy consumption in buildings could be another recommendation at this level of the study.”

Round 2
Reviewer 3 Report
The paper has been improved according to the comments.